# Low Cost, Ease-of-Access Fabrication of Microfluidic Devices Using Wet Paper Molds

**DOI:** 10.3390/mi13091408

**Published:** 2022-08-27

**Authors:** Raviraj Thakur, Gene Y. Fridman

**Affiliations:** 1Department of Otolaryngology, Head and Neck Surgery, Johns Hopkins University, Baltimore, MD 21205, USA; 2Department of Biomedical Engineering, Johns Hopkins University, Baltimore, MD 21205, USA; 3Department of Electrical and Computer Engineering, Johns Hopkins University, Baltimore, MD 21205, USA

**Keywords:** microfluidic devices, rapid prototyping, micromolding, polydimethylsiloxane (PDMS), microfluidics fabrication, microfabrication, xerography

## Abstract

Rapid prototyping methods enable the widespread adoption of microfluidic technologies by empowering end-users from non-engineering disciplines to make devices using processes that are rapid, simple and inexpensive. In this work, we developed a liquid molding technique to create silicone/PDMS microfluidic devices by replica molding. To construct a liquid mold, we use inexpensive adhesive-backed paper, an acetate backing sheet, and an off-the-shelf digital cutter to create paper molds, which we then wet with predetermined amounts of water. Due to the immiscibility of water and PDMS, mold patterns can be effectively transferred onto PDMS similarly to solid molds. We demonstrate the feasibility of these wet paper molds for the fabrication of PDMS microfluidic devices and assess the influence of various process parameters on device yield and quality. This method possesses some distinct benefits compared to conventional techniques such as photolithography and 3D printing. First, we demonstrate that the shape of a channel’s cross-section may be altered from rectangular to semicircular by merely modifying the wetting parameters. Second, we illustrate how electrical impedance can be utilized as a marker for inspecting mold quality and identifying defects in a non-invasive manner without using visual tools such as microscopes or cameras. As a proof-of-concept device, we created a microfluidic T-junction droplet generator to produce water droplets in mineral oil ranging in size from 1.2 µL to 75 µL. We feel that this technology is an excellent addition to the microfluidic rapid prototyping toolbox and will find several applications in biological research.

## 1. Introduction

Microfluidic technologies are revolutionizing numerous fields of science and have enabled several novel applications such as single-cell sequencing [1], organs-on-a-chip [2], and liquid biopsy [3]. At the heart of this technology are devices with microchannels that allow the controlled manipulation of liquid samples [4,5]. Microfluidic prototyping methods have played a central part in academic research. These methods are developed to make it accessible for non-engineering end-users such as biologists and life scientists. They are often optimized to have negligible process development times, making these methods a significant catalyst for the widespread adoption of microfluidic technologies. As of now, there are hundreds of publications on several different prototyping approaches relevant to microfluidics, spanning different substrate materials, channel sizes, device tolerances, production costs, and turnaround times [6,7].

For historical reasons and the prevalence of IC technologies, microfluidic devices were fabricated from silicon and glass using acid or plasma etching techniques. More recently, silicone-based elastomers, such as polydimethylsiloxane (PDMS), have been used extensively. This transition occurred because PDMS is easy to manufacture and is also optically transparent. The user-defined channels can be created by a straightforward casting procedure utilizing a master mold. After curing the polymer in a mold, the inverse pattern is transferred to PDMS. The polymer chip can then be peeled off from the mold and bonded to a featureless glass/PDMS sheet for sealing the device. When it comes to manufacturing microfluidic devices, designing and fabricating molds is a crucial first step.

Photolithography is the current gold standard for mold manufacturing [8]. This multistep method begins with the deposition of a thin coating of photoresist such as SU8 on a silicon wafer. The film is selectively exposed to ultraviolet (UV) light for crosslinking using a patterned photomask, and the unexposed portions are washed away using a developer solution. This protocol is well standardized, and the technique offers an outstanding resolution down to a few microns. However, it needs access to a cleanroom microfabrication facility, costly lithography equipment, and skilled personnel, all of which might be a barrier in a resource-limited environment. Additionally, the whole process is quite time-consuming, and it can take several days to create photomasks as this step is typically outsourced. As a result, there has been considerable interest from the microfluidic community in developing more accessible, quicker, and cost-effective ways for fabricating molds for PDMS devices. 

There are many examples of lithography-free methods described in the literature [9]. Broadly, they can be classified into three categories: 3D printing, print and peel (PAP), and traditional machining. Consumer-grade 3D printing technologies are continuously improving, both in quality and usefulness, and offer a user-friendly platform for mold manufacturing [10]. Fused deposition modeling (FDM) has been used, for example, to create PLA [11] or ABS molds for casting PDMS [12]. Numerous studies have also deployed stereolithography (SLA) [13] and digital light processing (DLP)-based 3D printing techniques to create resin-based master molds that are robust and reusable [14]. The advantages of this technique include quick turnaround times, resolution down to 100 µm, and the capacity to create multistep three-dimensional structures. However, recent studies have raised potential toxicity concerns because resin residues can diffuse into the PDMS device [15]. Three-dimensional printers were modified to print molds using biocompatible materials such as wax [16] and polyvinyl alcohol [17]. It is still an ongoing research topic to discover new materials and improve their print performance. Print and peel methods use solid-ink or LaserJet printers to directly print positive master on a transparent film [18]. This approach is limited to relief features depth in the range of 5–15 µmand lower aspect ratios [19]. Traditional machining approaches such as milling and laser ablation have also been shown viable in making acrylic and PDMS molds [20,21]. However, they may suffer from surface roughness and irregular channel topography [22]. Finally, digital cutters and laser engravers have also been used to make molds from adhesive tapes or laminates [23,24,25]. This approach is quite innovative and has proven to be as efficacious as some of the other techniques listed above.

Molds made using abovementioned methods are solid in nature. Here, we describe an innovative, liquid molding silicone/PDMS micromolding method that extends the capabilities of existing approaches by enabling features that are not possible with existing approaches (Figure 1). Our method only requires an off-the-shelf digital vinyl cutter and consumer-grade nitrocellulose paper. The procedure begins with creating paper molds via Xerography, custom patterning of the paper using the cutter, followed by wetting the mold with water via positive capillary action. We briefly introduced this method in our previous work to make a thin, flexible microfluidic nerve cuff device to stimulate and record sciatic nerve signals [26,27]. Here, we perform a more rigorous analysis by quantifying key performance indicators (KPIs) such as channel widths/depths, resolution, and surface roughness and determine the effect of key process variables such as water content and molding temperature. First, we show that both rectangular and round cross-section channels can be created by simply changing the water content of the mold, showing tremendous flexibility, compared to some of the other liquid molding methods [28,29]. Second, we demonstrate how simple electrical circuits can be created to detect mold defects in real-time by measuring changes in electrical impedance. This enables process quality control without relying on visual inspection. Finally, the molds are exempt from any surface treatment, making the process ideal for quick prototyping. We believe this method will equip life science and biology researchers with an unprecedented opportunity to make novel device architectures and will be a valuable addition to existing prototyping methods.

## 2. Materials and Methods

### 2.1. Manufacturing, Wetting, and Replica Molding Using Paper Molds

The entire end-to-end workflow is described in Figure 2. The following procedure has been utilized to produce paper-based master molds for replica molding. Avery Inc. (Marlboro, Md, USA) sticker paper was initially applied to a transparent acetate transparency film from Staples, Inc. (Framingham, MA, USA). The acetate paper served as a substrate for molds and provided a smooth background surface for peeling off the PDMS devices. Any visible trapped air bubble was removed using a smooth laminating roller (McMaster Carr, Inc., 7533A12, Elmhurst, IL, USA). The stacked sheets were then placed on the cutting mat of a digital vinyl cutter machine, Cricut Explore Air, Amazon, USA. The paper’s adhesive backing was discovered to be essential throughout the cutting procedure, as it served as an anchor for the cut pattern during the following molding process. The appropriate microfluidic channel shapes for cutting were created using INKSCAPE, a CAD tool, and the resultant pictures were transferred to the cutting machine to cut the paper selectively. The cutting of the acetate sheet was avoided by adjusting the cutting pressure parameter on the machine. The machine has several materials to select from *paper*, *light cardstock*, all the way to *fabric*. Each level generates a progressively higher value of cutting force, as described in the manufacturer’s technical sheet. From trial and error, we found that the *paper* setting was enough to cut through the sticky paper but not through the acetate film. After the cutting operation, the backdrop paper was carefully removed, leaving the paper molds on the transparent acetate backing.

As the next step in the molding process, the dry paper molds fabricated using the above steps were wetted by pipetting a known amount of water onto the mold surface using a handheld pipette (P200, Gilson, Inc., Middleton, WI, USA). Due to the positive capillary action, the paper is wetted. The pipetted drop was placed on the reservoir well part of the mold. We used both deionized water as well as electrically conducting saltwater (NaCl, 0.9 wt%) for this process. No appreciable difference was found between salt water and deionized water later on for the molding quality. The amount of water used was a rather critical parameter determining the success of the process as well as affecting the geometry of the molded channels. The total dry paper mold volume was first calculated in µL by assuming a paper depth of 100 µm (corresponding to a 4 mil paper sheet). Then, depending on the desired water fraction, the total volume of water was calculated by multiplying the water fraction by the volume of the dry paper mold. The water fraction changed depending on the experimental conditions under investigation.

We utilized two different types of silicones in this investigation. One was a quick cure silicone that cured in 10–20 min at room temperature (Ecoflex 00-35 Fast Platinum Cure Silicone Rubber Compound Kit, Smooth-On Inc., Macungie, PA, USA). The second was polydimethylsiloxane, which was purchased from Ellsworth Adhesives (Sylgard 184 kit) and cured within 12–36 h at temperatures depending on the experimental parameters. After wetting the molds, the pre-polymer liquid was gently poured over them and incubated in a temperature-controlled oven. Once the polymer was cured, the acetate film along with the paper mold was peeled off from the silicone/PDMS chip. Notice that this step is inverse to existing solid molds, where the final PDMS chip is peeled off from the mold. Finally, to make an enclosed chip, featureless PDMS slabs were plasma bonded to the molded layer. Food colors dissolved in water were used for visualizing prototype microfluidic PDMS devices, as seen in Figure 3c. 

### 2.2. Measurement of Channel Width, Depth, Surface Roughness, Wall Angle, and Curvature Index

A laser scanning microscope (Keyence, Itasca, IL, USA) was used, configured for reflecting surfaces, to take brightfield images as well as for three-dimensional surface scanning of the mold and the channel. An in-built program was used to quantify surface roughness. The z-axis scans were limited to a ±100 µm range to limit outliers and artifacts from surface reflection. Cured microfluidic devices were cut through an axis perpendicular to the channel and imaged using a boom microscope and a USB camera. The ImageJ program was used to measure channel depth, width, side-wall angle and perimeter after setting up a scale (pixel/distance) for individual images.

### 2.3. Electrical Impedance Measurement Setup

Hypodermic 16-gauge stainless steel needles were used as electrodes. The electrodes were embedded in the reservoirs of the mold layout. Electrical impedance was measured using an impedance analyzer, Cypher Instruments, C-60 (LDN, UK). The impedance was measured from 10 Hz to 10 kHz to fully characterize the electrochemical behavior of the circuit. The impedance-frequency data was exported into CSV files for further visualization and analysis.

### 2.4. Statistics and Data Visualization

Six microfluidic device replicates were made using each of the data points in Figure 4, Figure 5 and Figure 6 to account for experimental uncertainty. The points represent the mean of the measured values, whereas the error bars represent ±2 * standard deviations to encompass a 95% confidence interval. Graphpad Prism (Version 7.0d, GraphPad Software, Inc., San Diego, CA, USA) was used for the calculation of errors and line fitting as well as for data visualization using scattered plots. Unpaired *t*-test was performed between different groups of each control variable to calculate statistical significance. We assumed gaussian distribution throughout.

### 2.5. Water-in-Oil Emulsion Generator Chip

A CAD design was first created to make the droplet generator chip. It had two inlet reservoirs and one outlet reservoir. The inlet channels, T junction, and outlet channel were all designed to be 500 µm in width. After the xerography process, the mold was wetted with 4.5 µL of water, and fast-curing silicone was used to make this device. The curing time was approximately 20 min. Then, the mold was peeled off, and holes were punched in the reservoir to press-fit 1/16 OD silicone tubing (Tygon, Mcmaster Carr, Elmhurst, IL, USA). The chip was sealed using a featureless poly-dimethylsiloxane (PDMS) featureless base using corona treatment, as described previously. Three tubes were then press-fitted, one for mineral oil, one for water, and one to collect the generated droplets. Each tubing connected the inlet of the droplet generator to a 25 mL plastic tube fluid reservoir, held in position using height-adjustable lab clamps. For the droplet generator experiments, the dispersed phase (miner oil) was held at +140 mm height, and the aqueous phase (water) was held at 80 mm with respect to the droplet generation chip, which is used as datum here (z = 0 mm). The images of the droplets were recorded using a boom microscope and a USB camera.

## 3. Results and Discussion

The mechanism of the proposed micromolding process is simple and can be explained by the principles of capillary action and liquid partitioning. Figure 1a shows different paper patterns placed on transparency film that was designed using a CAD program and cut using the digital cutter. As examples, we chose mold geometries that are routinely used in microfluidic devices, namely T/Y/cross junctions, spirals, zig-zag channels, and square microwell array. Figure 1a also shows the wetting process of the mold, where water, stained with blue food color, was added to the mold. Due to positive capillary suction, the liquid quickly spreads across the paper. A hydrophobic bottom substrate of acetate paper is rather necessary to ensure that wetting is confined within the boundaries of the porous paper pattern. The partition coefficient between water and silicone is negligible, and when the pre-polymer mix is poured over the wetted mold, the liquid mold remains intact. A certain threshold of water content is needed for the entire coverage of the paper pattern and effective molding to occur. On the other hand, if pre-polymer mix is added without the addition of water, the polymer enters the pores of the paper. In this case, no molding can occur, and the paper pattern becomes entrapped during the curing process, as evidenced during the peeling off operation. This hypothesis was confirmed using a negative process control, as shown in Figure 1b. Figure 2 shows the entire workflow of the micromolding process, which is described in the Materials and Methods section. Figure 3 shows examples of resultant microfluidic devices made in PDMS. The channels are primed with food color for ease of visualization.

Next, we performed a characterization of channel widths and depths to determine the resolution of this process. A series of straight rectangular channels were designed in the CAD program with varying widths, starting from 150 µm and ending at 1 mm in the steps of 50 µm. Figure 4a shows the images of paper molds and corresponding PDMS microchannels obtained using bright field mode. Figure 4b shows a three-dimensional scan of mold and cast microchannel having a specified width of 250 µm. The measured channel widths for both paper molds, as well as casted PDMS channels, are plotted in Figure 4d,e, respectively. Six replicate devices were made using a new cutting blade each time to account for experimental uncertainty. As seen from the plots, the molds appeared to be smaller than the specified CAD program width, whereas casted channels appeared to be larger. A negative error during cutting operations can be explained by the loss of material. A linear fit in the data with a system error of −25.42 µm showed a high R-square value of 0.98, as seen in Figure 4d. On the other hand, the casted channel appeared larger in size than the mold. We later concluded that the paper mold expands upon the absorption of water, resulting in higher than specific channel dimensions. In this scenario, a linear fit with a positive error of 0.3 µm showed a high R-square value of 0.9863. We found the smallest mold width that can be successfully cut into the sticky paper to be 175 microns. The blade sharpness (old vs. new), geometry (30, 45, 60 blade angle), and paper type setting in the digital cutter had a significant impact on the cutting resolution (data not shown). We found the best practice was to replace the blade after a few hours of cutting, which ensured a cutting resolution of 175 µm. The sticky paper used for molds had a thickness of 4 mil (100 µm). Figure 4e shows that deeper channels can be created by stacking multiple prior to the cut. We achieved a depth of ~500 µm by stacking three sheets. From the data, it is also evident that the depth of the resulting channel was more than the mold itself, indicating the expansion of paper z-direction from upon the absorption of water. This behavior is amplified as more layers of paper are stacked on top of each other, which is seen in Figure 4f.

An important question related to this process is how much water should be added to the paper mold and how it affects the overall fabrication quality. To quantify this, we first define the term water fraction, which is equal to the ratio of the volume of added water (µL) to the volume of the paper mold pattern. For a simplified analysis, we created a series of straight channel molds 2 cm in length and 2 mm in width. With 4 mil paper having a depth of ~100 µm, the volume of this mold was calculated to be ~2 µL. We varied water fractions from 0 (dry) all the way to 3.5 (oversaturated). The water fraction determined the overall mold morphology. For example, a water fraction of 0.5 could not wet the entire surface area of the paper, whereas a water fraction of 1 was found to have just enough coverage. A water fraction of 3.5 resulted in a visual convex meniscus profile, as seen in Figure 5b. We measured the effect of water fraction on six quality parameters-% change in width, % change in depth, wall angle, curvature index, and surface roughness parameters Ra and Rq, as seen in Figure 5c–h. Representative images of the channel’s top view, laser greyscale scan, surface, and cross-sectional profile for each of those experimental conditions are shown in Figure 5a. It was observed that a minimum of 0.75 was needed to yield a successful process. Below this threshold, the paper was trapped fully or partially into the silicone chip, as seen in Figure 5a. Both channel depth and width increased with increasing water fraction. Expectedly, the change in width was ~20% for a water fraction of 5, whereas the change in depth was almost 400% with respect to the specified dimension (Figure 5c,d). This confirmed our hypothesis that water added remains largely confined by the boundaries of the pattern. The negative percent change in channel depth in Figure 5d indicates that the paper was fully or partially trapped in the PDMS/ecoflex channel after the molding process. Thus, measured depth is smaller than the specified depth, resulting in a negative percentage change. This happens if the paper mold is either completely dry or partially dry, which leads to polymer infiltration into the paper. We also measured two important profiling parameters, i.e., wall angle and curvature index, as shown in Figure 5e,f. Wall angle is defined as the angle between the side wall and channel base. Wall angle was found to increase with increasing water fraction. The minimum angle was around 90 degrees implying near-vertical walls, while the maximum was measured to be plateaued around 140 degrees at the water fraction of 2.5. The curvature index, on the other hand, is defined as 4×Π×(area)/(perimeter)^2. As an example, a perfect circle has a curvature index of 1. It can be seen that from Figure 5a,c, the cross-sectional shape transformed from a rectangle with a curvature index of 0.2 to a semicircular with a curvature index of 0.5. As water fraction increases, the effect of surface tension starts to play a dominating role with meniscus formation commencing over the paper mold. This is a desired feature of this manufacturing technique that would enable rapid fabrication of circular microchannels with ease like no other rapid prototyping method. Next, we measured surface roughness parameters, since it is an important metric to determine device quality. Surface roughness is a key parameter that will affect device performance in certain applications. An example of such an application is monolayer of endothelial cells inside a microfluidic channel, which is often used in organ-on-chip assays. If the surface is rough, it directly affects the adhesion and distribution of cells on the surface. Instead of a monolayer and uniform coverage, cellular aggregates are formed. Our follow up publication will focus on this aspect more and how we utilize the proposed micromolding method to form thin blood capillaries. A rough surface can result from the trapping of cellulose fibers, as well as potential bubbles trapped from the evaporation of water during the curing process. Both average (Ra) and root mean squared (Rq) of profile height deviations are plotted as a function of water fraction in Figure 5c. There appears to be a systemic error in the measurements, given that the acetate sheet has a surface roughness of <100 nm and included Ra and Rq values for the acetate sheet in the plot as a control. It can be concluded that with increasing the water fraction of the mold, the channels become smoother. Surface roughness was quite high for water fractions less than 0.75, with measured values of as high as ~30 µm for Ra and Rq, indicating that in this region, the majority of cellulose fibers are trapped during the molding process. At a water fraction of 1.5, channels were almost as smooth as the substrate, with Ra and Rq around 1–2 µm. As the water fraction increased, it was challenging to correct for the curviness of the channel, and thus we firmly believe that the actual values are significantly lower than the measured values reported here. In Figure 5c, we have drawn dotted lines on each plot to indicate a threshold value for water fraction. It is important to note that for experiments covering Figure 5, we used fast-curing silicone as an analog to PDMS due to its low curing time. There was no difference observed between the two polymers in terms of molding (data not shown). Along with water fraction, another important process parameter was curing temperature. In the case of solid molds, a general curing temperature for PDMS can range from 60 to 120 °C. To measure the effect of curing temperature on the molding process, we made PDMS chips of a simple microfluidic channel and examined them visually under a microscope. For curing temperatures above 45 °C, the process yielded devices with large bubble defects, as seen in Figure 6a. Below 45 °C, the casted channels were smooth and defect-free. For this protocol, our recommended curing temperature is between 30 and 45 °C for PDMS.

Quality inspection still remains one of the biggest challenges in rapid microfluidic prototyping. In the case of existing solid molds, the only method to detect defects is via visual inspection using a microscope. More often, the real defect is detected after a device is made by observing unusual behavior such as leaking or clogging. On the contrary, we show how our molding method permits simple electrical impedance checks to monitor mold quality and detect any post-curing defects. An example microfluidic device is shown in Figure 6b,c, which has a defect. By design, two parallel channel molds have a finite gap between them, but because of an arbitrary defect, the liquid from the mold is leaked between the two, causing the molds to merge. When the two molds were inspected under a microscope, no difference was observed under visual inspection, as shown in Figure 6c. The defect was only confirmed later, once the device was fully assembled and food colors were used, as seen in Figure 6d. Next, a two-electrode impedance spectroscopy setup was used instead of a visual inspection to measure the impedance across the two molds, Figure 6e. Here, sodium chloride was added to the water to make it electrically conductive. By measuring the impedance jump, this defect can be easily detected at any stage during device manufacturing. For experimental purposes, defects were introduced using two predetermined methods. In the first method, water fraction was increased in each of the two adjacent channel molds with a hypothesis that the two menisci would come in contact with each other causing them to merge and giving rise to an undesired channel defect. In the second method, a defect was introduced by spinning the substrate inside of a thin film coating machine. Figure 6f,g show the measured impedance across the two-electrode setup at 10 kHz as a function of water fraction and the revolution speed measured in rpm, respectively. At a threshold water fraction value of 3.5, an impedance jump was noticed, implying that the defect occurred. Similarly, at the threshold speed of 2000 rpm, an impedance jump was also observed, implying the two liquid molds merged possibly because of centrifugal forces. Additionally, Figure 6h shows an example of the frequency-impedance spectra for a mold geometry plotted for different water fractions. Such a characterization can be used as a calibration curve to determine the amount of water added to the mold, providing additional methods to check on mold quality.

In our previous work, we demonstrated how this technique can be used to create a thin, flexible, special-purpose device called microfluidic nerve cuff [26,27], with a fabrication protocol that will not be possible with solid molding techniques. We also showed that such a device can be used to stimulate as well as record from the peripheral sciatic nerve. Here, as a second example, we made a microfluidic droplet generator chip. The chip CAD design and images of the molding process have been shown in Figure 7a,b. A gravitational pressure head was used as a fluid drive for both mineral oil and water (mixed with green food color) together through the T junction of the chip to make a water-in-oil emulsion, as shown in Figure 7c. The T junction was designed to be 500 µm in width, 500 µm in-depth, and 100 µm in height. By simply adjusting the relative height difference between the two reservoir columns, droplets of varying size were generated, as seen in Figure 7d. The smallest droplet was around ~1.7 µL, whereas the largest droplet was ~78 µL.

Finally, we would like to include some design considerations for the wet paper-based silicone/PDMS micromolding technique presented in this paper. Table 1 shows a comparative chart of the proposed method along with existing microfluidic prototyping methods. (1) As described in the earlier section, the resolution of this technique is ~175 µm. Minimum width and depth achievable with this method are 175 µm and 110 µm, respectively. The depth is controlled by the thickness of the sticky paper itself, while the width is dependent on the resolution of the digital cutter. We found that the minimum gap spacing achievable is also the same as the minimum width, 175 µm. However, as seen in Figure 6f, this metric is indeed dependent on the water fraction. Increasing water fraction will likely reduce the resolution because water menisci from adjacent molds are likely to merge, lowering the resolution. (2) Given the nature of this technique, more complicated liquid architectural shapes can be formed. As an example, the reservoir design can include concentric rings/squares shape, which will allow a larger droplet to anchor. Additionally, this technique can be used to make rounded channels, a feature that will have several uses in organs on-chip applications for mimicking blood vessels. (3) The best curing temperature range for PDMS is 30–45 °C. This will ensure chip fabrication without any issue of bubble defects from the evaporation of water. (4) When choosing a sticker paper, care must be taken to ensure that the adhesive is not water-soluble as it can negatively impact the process. (5) Impurities in the form of dust or paper fibers from cutting operation can accumulate on the acetate substrate, potentially changing wettability properties. In such cases, added water could spread past the boundaries of the paper pattern. For future studies, we will deploy a substrate cleaning step as well as extensive characterization for different backing adhesives. (6) The surface roughness of the acetate sheet was not provided by the sheet manufacturer. However, it is safe to assume that they have a mirror-like surface finish since these sheets are manufactured by thermal extrusion using roll to roll manufacturing. Additionally, we bonded the droplet generator chip with a blank PDMS slab to enclose the open channels and any surface roughness from the acetate sheet would have caused leakage or a lack of bonding. While we did not quantify the adhesion forces between wet paper mold and acetate film, adhesion was always strong enough to withhold the peeling-off operation. Throughout our prototyping experiments, we did not observe the diffusion of PDMS/ecoflex in between the wet paper and acetate sheet. However, the wetting properties of the glue can be an important factor which may cause such a malfunction. We have not explored this experimentally, mainly because finding different adhesive backed paper off-the-shelf is challenging. (7) Surface tension plays a significant role in oversaturated wet paper mold, resulting in rounded PDMS channels. The shape will directly depend on the surface tension values between water and PDMS/silicone and could potentially be altered by the use of an appropriate surfactant.

One particular disadvantage of this method compared to existing rapid prototyping approaches is that curing time is higher (12–36 h) than existing solid molding methods such as photolithography or 3D printing (0.5–4 h). A potential solution to this problem that we are investigating is either using mineral oils or polymeric solutions that are immiscible with silicone/PDMS. Our preliminary experiments using mineral oil have been successful (data not shown), but a more comprehensive study is needed. Another limitation of this technique could be sharp edges/corners mainly for rounded channels. The permissible angle will be determined by the surface tension dynamics for oversaturated wet mold and the cross-sectional profile could be altered. While Liquid Molding is a promising solution for flat geometries, designs that require step or incremental z-direction changes are more challenging, as they would require stacking multiple layers of paper in different parts of the mold. A more thorough study is needed to address the limitations of channel shapes and more complicated device topology.

## 4. Conclusions

We have successfully developed a novel liquid molding method for making microfluidic devices in PDMS. Our method simply requires a digital cutter to pattern paper, which, when wet, can be used as a master mold due to the immiscibility of water and the silicone polymer family. After extensive characterization, we found that a minimum rectangular channel cross section possible with this technique is 175 µm (width) × 120 µm (height). Furthermore, a water fraction of approximately 1–1.25 is needed for a smooth surface finish. We also found that higher curing temperature negatively affects the molding process, and the ideal temperature of the proposed method is in the range of 30–45 °C. We highlight both the advantages and disadvantages of this method compared to current processes and also suggest design considerations that will help researchers who want to deploy this method. A detailed study of limitations in terms of device architecture and shapes is currently under investigation.

## Figures and Tables

**Figure 1 micromachines-13-01408-f001:**
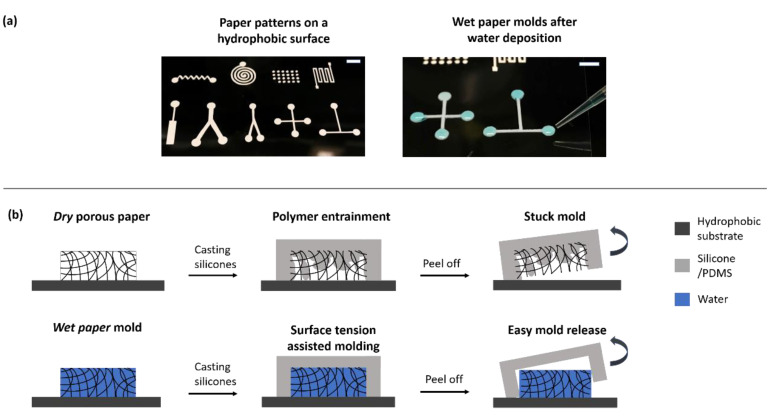
Replica molding of silicone/PDMS using wet paper molds. (**a**) Images showing the paper mold shapes on a transparent hydrophobic transparency paper, cut using a digital vinyl cutter. The second image shows the wet molds resulting from pipetting of appropriate amount of water followed by capillary suction from the porous paper. The scale bar for both images is 5 mm. (**b**) Conceptual schematic showing the liquid-assisted molding process. When a dry paper is used as a master mold, polymer premix enters its pores causing the mold to stuck on its surface. On the other hand, after wetting the paper, water infiltrates all the pores of the porous paper matrix. Since water is immiscible with silicones, addition of water facilitates an easier peeling off when the polymer is cured.

**Figure 2 micromachines-13-01408-f002:**
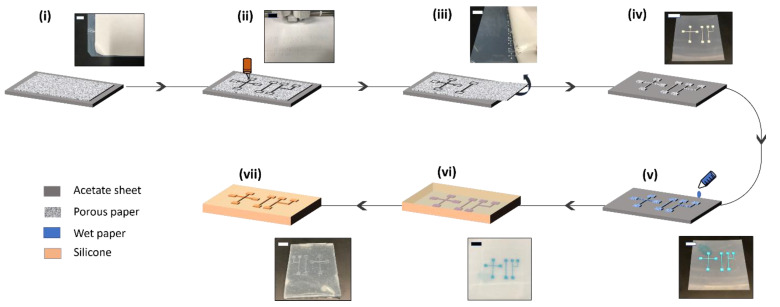
Fabrication workflow of the wet paper molding process. The mold fabrication only requires a cellulose paper sheet, an acetate sheet and a vinyl cutter. A sticky paper is laid on an acetate sheet (**i**) and a digital vinyl cutter is used to cut the mold shapes (**ii**). After carefully peeling off the background (**iii**), the dry molds are inspected for desired cutting operation (**iv**). Measured aliquots of water are pipetted on the molds and allowed to wet the paper through capillary action (**v**). Polymer mix is added on top (**vi**) and after curing, the chips is pulled from the mold (**vii**). The scale bars represent 1.5 cm for (**i**–**iii**) and 5 mm for (**iv**–**vii**).

**Figure 3 micromachines-13-01408-f003:**
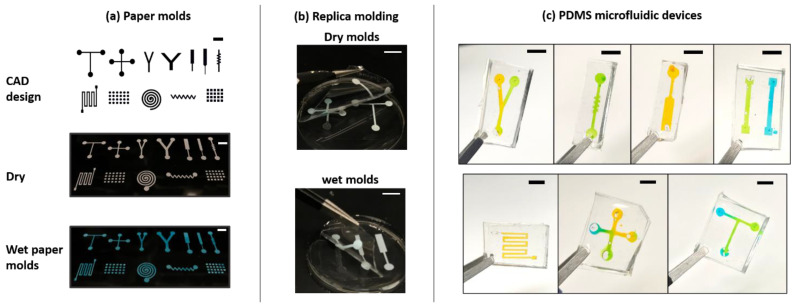
(**a**) Images showing CAD drawings and the corresponding dry and wet paper molds manufactured using the fabrication workflow. (**b**) Images showing that when the mold is dry or the amount of water added is not adequate enough, the molding process fails and the paper is stuck on the cured polymer, whereas adding the right amount of water results in a successful replica molding and the mold can be peeled off from the cured polymer layer. (**c**) Images showing the PDMS chips with various standard geometries fabricated from the wet molds. The scale bar for each image is 5 mm.

**Figure 4 micromachines-13-01408-f004:**
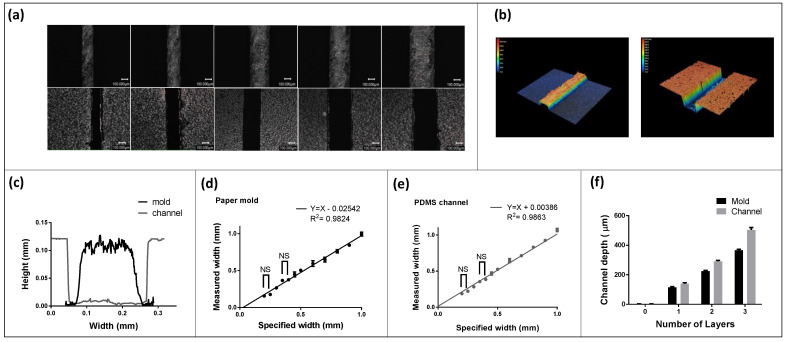
Characterization of liquid based micromolding process. (**a**) Several brightfield microscopic images showing the mold and the casted channel for different widths. The smallest channel width that could be replicated reliably was found to be ~175 microns. The scale bar represents 100 µm on each image. (**b**) 3D laser scan showing the paper mold and the corresponding molded microchannel in silicone. (**c**) Cross-sectional profile of the mold and the channel showing that the channel width is dependent on the amount of water added. (**d**) Plot showing measured width vs. specified width for the paper mold. A positive systemic error results from the cutting operation where the material is lost. (**e**) Plot showing measured width vs. specified width for the casted PDMS microfluidic channel. NS represent non-significant groups for which *p* > 0.05 whereas rest of the measurements were found statistically significant based on unpaired *t*-test and yielded a *p* < 0.001. (**f**) The channel depth can be controlled by stacking more paper layers.

**Figure 5 micromachines-13-01408-f005:**
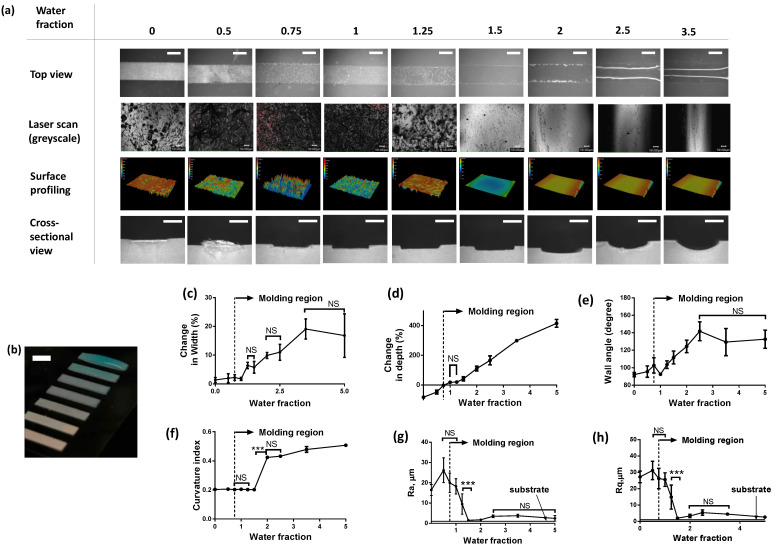
Effect of mold water content on the replica molding process. (**a**) Images and laser scans showing the effect of progressively increasing the water fraction of a mold. A minimum water fraction of 0.75 was needed to commence the molding process. The scale bars for top view and cross-sectional images are 1 mm. The scale bar for the laser scan grey scale images is 100 µm. (**b**) Image showing how the dry paper mold appears upon progressive addition of water. The scale bar on the image is equivalent to 2 mm. Even beyond the saturation point, the excess water is retained on the mold since the bottom substrate is hydrophobic. Characterization plots can be seen showing the effect of increasing water fraction on the channel width (**c**), on the channel depth (**d**), the wall angel (**e**) and curvature index (**f**). Additionally, the effect of water fraction on the resulting channel roughness was evaluated by measuring the gold standard roughness parameters, Ra and Rq as seen in (**g**) and (**h**). NS represent non-significant groups for which *p* > 0.05 whereas rest of the measurements were found statistically significant based on unpaired *t*-test and yielded a *p* < 0.001 denoted by ***.

**Figure 6 micromachines-13-01408-f006:**
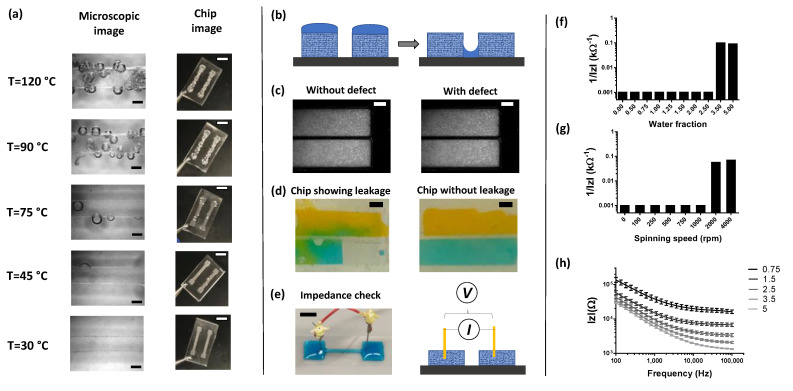
Effect of curing temperature and electrical impedance-based mold defect detection. (**a**) Several photos of PDMS chip cured at different temperature. It can be seen that at higher temperatures, the water from the mold evaporates, resulting in bubbling in the polymer solution. The scale bars represent 1 mm distance for microscope images and 5 mm for device photos. (**b**) Schematic showing a defect where the two channels which are separate get connected because of coalescence of water. Such defect is not visible as seen in (**c**) and can be seen in the fabrication chips after leakage test (**d**). Liquid molding offers a distinct advantage in detecting such defects by using impedance measurements (**e**) by using a two-electrode setup. A defect such as discontinuity can be detected by observing the jump in impedance profiles. (**f**) An example of a defect in the mold caused because of coalescence detected pre-molding process. (**g**) An example of a defect in the mold post molding operation caused by coalescence due to spinning the mold during the molding step. (**h**) Impedance-frequency spectra for various water fractions of the mold showing such curve can be used to create calibration plot for knowing the amount of water added in the mold. All the scale bars represent 1 mm distance for images in (**c**–**e**).

**Figure 7 micromachines-13-01408-f007:**
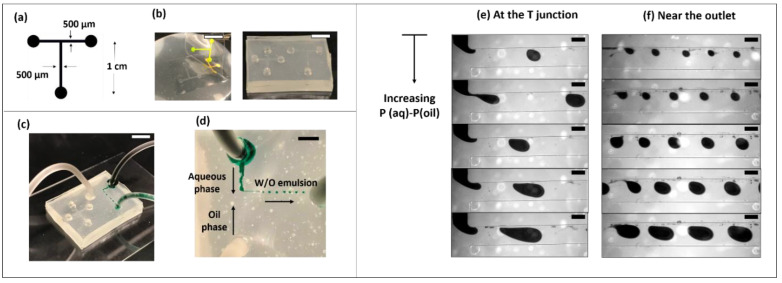
A water-in-oil emulsion generator microchip. (**a**) A CAD input file to the vinyl cutter showing the dimensions of the T junction. (**b**) The resulting silicone microfluidic chip from the wet paper mold. (**c**) Image showing the experimental set up where two fluids were flowed into the chip using gravity driven flow. (**d**) A zoomed-in view of the T junction showing the formation of water in mineral oil emulsions. Pressure difference between the two inlets was varied by changing the column height. Images showing microscopic view at the junction (**d**) and near at the outlet (**e**) can be seen where droplet sizes increase as a function of increasing pressure difference. For this demonstration, the column height of oil was maintained at the same level whereas the aqueous inlet column height was varied to modulate the pressure differential. The scale bars represent 5 mm for images in (**b**–**d**), while for all the images in (**e**,**f**), it is 250 µm.

**Table 1 micromachines-13-01408-t001:** Comparison of the proposed liquid micromolding with other microfluidic prototyping methods.

Method	Cost/Mold	Capital Equipment and Cost	Resolution (Width × Depth)	Turnaround Time	Advantages	Limitations
Photolithography	>$100	UV Mask aligner, >$10k	10 µm × 0.5 µm	3–5 days	High resolution	Depths > 100 µm, Round cross-sections
Additive Manufacturing	<$1	3D printers ($1000)	100 µm × 50 µm	<1 day	Fast	Uncured resin diffusion into PDMS
Subtractive Manufacturing	>$10	CNC milling ($5k)	10 µm × 1 µm	<1 day	Large footprint	Surface roughness
Liquid Molding	<$1	Digital cutter ($100)	175 µm × 110 µm	<1 day	Round Cross section	Sharp corners/turns

## Data Availability

Not applicable.

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
