# Peer review of "Low Cost, Ease-of-Access Fabrication of Microfluidic Devices Using Wet Paper Molds"

_micromachines, 2022, doi:10.3390/mi13091408_

Round 1

Reviewer 1 Report

In this paper, Thakur et al. present a liquid molding technique to fabricate PDMS microfluidic devices using adhesive backed paper stuck to an acetate baking sheet. Using a vinyl cutter they demonstrate the ability to fabricate channels as small as 175 um. They show that wetting the paper mold with water improves the surface texture of the fabricated channels. Additionally, they demonstrate that measuring the impedance of the channel offers an easy way to perform a quality control of the fabricated mold. Finally, they use the method to fabricate a microfluidic droplet generator.

This method is very accessible and allows for the rapid prototyping of microfluidic devices without the need for expensive photolithography equipment. This would be a very useful method for practitioners in microfluidics. The authors do a thorough job of analyzing the effect of different process parameters on the quality of the devices and also discussing the advantages and disadvantages of their method. I recommend the publication of this manuscript.

Some minor comments for revision:

  • Line 136 - Should it be Sylgard 184?

  • In Fig 4 d/e, please mention which plot is for the paper mold and which one is for the fabricated microfluidic channel. This is mentioned in the discussion, but mentioning it in the caption will make it easy for the reader to understand the figure.

Author Response

Reviewer 1 In this paper, Thakur et al. present a liquid molding technique to fabricate PDMS microfluidic devices using adhesive backed paper stuck to an acetate baking sheet. Using a vinyl cutter they demonstrate the ability to fabricate channels as small as 175 um. They show that wetting the paper mold with water improves the surface texture of the fabricated channels. Additionally, they demonstrate that measuring the impedance of the channel offers an easy way to perform a quality control of the fabricated mold. Finally, they use the method to fabricate a microfluidic droplet generator. This method is very accessible and allows for the rapid prototyping of microfluidic devices without the need for expensive photolithography equipment. This would be a very useful method for practitioners in microfluidics. The authors do a thorough job of analyzing the effect of different process parameters on the quality of the devices and also discussing the advantages and disadvantages of their method. I recommend the publication of this manuscript.

Thank you.

Some minor comments for revision:
• Line 136 - Should it be Sylgard 184?
Thanks for noticing this. We fixed the typo.
• In Fig 4 d/e, please mention which plot is for the paper mold and which one is for the fabricated microfluidic channel. This is mentioned in the discussion, but mentioning it in the caption will make it easy for the reader to understand the figure.
We revised the caption and introduced the distinction into the plots.

Reviewer 2 Report

The authors Raviraj Thakur et.al., have proposed an interesting approach towards fabrication of PDMS based microfluidic devices. The proposed method is a liquid molding process and might be a faster and cheaper solution towards fabrication of PDMS devices with certain size limitations. 

Following are my point by point comments.

  1. The authors should include a comparative chart of different molding methods and how they compare with the proposed method.
  2. What are the limitations of the proposed method? Authors are suggested to include this in discussion.
  3. Figure 4d and 4e looks identical. Also the figure caption is the same. What is the difference between these two figures?
  4. What is minimum achievable feature size with this method? (Eg. minimum width, depth, gap spacing?)
  5. As seen from Figure 4f, increasing the number of layers increases the route between mold depth and Chanel depth. Why is it so?
  6. In figure 5d, why percent change in depth is negative for water fraction less than 1?
  7. The authors should include a section for explaining the significance of all the graphs mentioned in Figure 5 just after the the appearance of Figure 5 for better understanding to the readers.
  8. The authors should indicate p values for all the graphs.
  9. How does surface tension plays a role in the whole fabrication process. The authors might include a short description.
  10. Have the authors performed the fabrication process using different thickness of acetate paper? 
  11. Although authors have demonstrated the application of the proposed fabrication method with droplet generator device, readers might be interested to know of complex geometries can be fabricated using proposed method or not?

Author Response

Reviewer 2

The authors Raviraj Thakur et.al., have proposed an interesting approach towards fabrication of PDMS based microfluidic devices. The proposed method is a liquid molding process and might be a faster and cheaper solution towards fabrication of PDMS devices with certain size limitations. 

Following are my point by point comments.

  1. The authors should include a comparative chart of different molding methods and how they compare with the proposed method.

Thank you for that idea.  We added Table 1 to highlight the features of the proposed method with existing prototyping approaches.

  1. What are the limitations of the proposed method? Authors are suggested to include this in discussion.

We added additional observations on this topic to the Results and Discussion. One major limitation of this technique could be sharp edges/corners mainly for rounded channels. The permissible angle will be determined by the surface tension dynamics for oversaturated wet mold and cross-sectional profile could be altered. Secondly, creating a step profile cross-section can provide optimization challenges in terms of cutting force parameters when dealing with 2 layers of stacked paper molds.  

  1. Figure 4d and 4e looks identical. Also the figure caption is the same. What is the difference between these two figures?

Fig 4d is the measured width of the paper mold whereas 4e is the measured width of the resulting microchannel. We corrected the caption and added a color scheme to agree with 4f for better visualization.

  1. What is minimum achievable feature size with this method? (Eg. minimum width, depth, gap spacing?)

Minimum width and depth achievable with this method are 175µm and 110 µm respectively. The depth is controlled by the thickness of the sticky paper itself while the width is dependent on the resolution of digital cutter. We found that minimum gap spacing achievable is also same as minimum width, 175µm. We have added this information in the results and discussion as well as to the new comparison Table.

  1. As seen from Figure 4f, increasing the number of layers increases the route between mold depth and Chanel depth. Why is it so?

The primary reason for this behavior is that there is expansion of the paper in z direction from water absorption. This behavior is amplified as more layers of paper are stacked on top of each other which is seen in Fig 4f.

We improved the explanation in the Results.

  1. In figure 5d, why percent change in depth is negative for water fraction less than 1?

The negative percent change indicate that the paper was fully or partially trapped in the PDMS/ecoflex channel after molding process. Thus, measured depth is smaller than the specified depth resulting in negative percentage change. This happens if the paper mold is either completely dry or partially dry which leads to polymer infiltration into the paper. We added this discussion to the manuscript.

  1. The authors should include a section for explaining the significance of all the graphs mentioned in Figure 5 just after the appearance of Figure 5 for better understanding to the readers.

We rearranged the figures to match the corresponding text.

  1. The authors should indicate p values for all the graphs.

Thank for noticing this unintended omission. We calculated p values for all the plots using unpaired t-test between different groups and assuming gaussian distribution throughout. Our calculations indicated p<0.001 for most of the measurements. The data where P values were >0.05 are indicated in the plot with label ‘NS’. We added notes in the methods section to include statistical significance calculations and updated figure captions to include statistical results. Plots in Fig 6 are real-time, single event measurements of a random mold defect and thus, are exempt from any statistical significance.

  1. How does surface tension plays a role in the whole fabrication process. The authors might include a short description.

Surface tension plays a significant role in oversaturated wet paper mold, resulting in rounded PDMS channels. The shape will directly depend on the surface tension values between water and PDMS/silicone and could potentially be altered by use of an appropriate surfactant. This parameter is not included in the current experimental design mainly due to scarcity of surfactants for PDMS and silicone family in general. We added this explanation to the result and discussion section.

  1. Have the authors performed the fabrication process using different thickness of acetate paper? 

The acetate paper thickness does not play a role since it is only used as a backing substrate over which paper is patterned and thus this was not included in the current experimental design.

  1. Although authors have demonstrated the application of the proposed fabrication method with droplet generator device, readers might be interested to know of complex geometries can be fabricated using proposed method or not?

The main limitation of complex geometries is differential depth of z-direction in the mold. We added the explanation to the final paragraph of results and discussion section.

Reviewer 3 Report

This manuscript by Thakur et al. described an interesting approach to make microfluidic device from wet paper mold. The authors investigate the influence of different parameters on the molding results and demonstrated a water-in-oil emulsion generator using their method. This manuscript is generally well-written and the results sound. Recommend to publish after addressing the following issues.

1. In the introduction, the authors claim that the photolithography process for fabricating microfluidic channel takes several days, which may not be true. The spin-coating of SU-8 and exposure/develop only takes several hours at most.

2. When the author using the Vinyl cutter to cut the cellulose paper, how to avoid cut the acetate sheet?

3. What's the surface roughness of the acetate sheet? How strong is the adhesion between acetate sheet with the cellulose paper? Will the PDMS/Ecoflex liquid diffuse into the interface between acetate sheet and cellulose paper?

4. How the surface roughness of molded microfluidic channel influence the liquid inside?

5. Indicate the unit of Ra and Rq in Figure 5g and h.

6. Figure 1 in page 8 should be Figure 7 and Figure 7 in page 11 should be Figure 8.

Author Response

Reviewer 3

This manuscript by Thakur et al. described an interesting approach to make microfluidic device from wet paper mold. The authors investigate the influence of different parameters on the molding results and demonstrated a water-in-oil emulsion generator using their method. This manuscript is generally well-written and the results sound. Recommend to publish after addressing the following issues.

Thank you.

  1. In the introduction, the authors claim that the photolithography process for fabricating microfluidic channel takes several days, which may not be true. The spin-coating of SU-8 and exposure/develop only takes several hours at most.

We agree that the development and exposure is not a long process. However, while spin coating of SU-8 and the soft lithography process may only take a few hours, the photomasks need to be either printed on a transparency or chrome etched on a glass substrate from a specialized vendor. This timeline normally takes several days from the inception of the microfluidic design.

  1. When the author using the Vinyl cutter to cut the cellulose paper, how to avoid cut the acetate sheet?

We avoided the cutting of acetate sheet by adjusting the cutting pressure parameter on the machine. The machine has several materials to be selected, from paper, light cardstock, all the way to fabric. Each level generates a progressively higher value of cutting force as described in the manufacturer’s technical sheet. From trial and error, we found that paper setting was enough to cut through the sticky paper but not through the acetate film. We added this information in the experimental section.

  1. What's the surface roughness of the acetate sheet? How strong is the adhesion between acetate sheet with the cellulose paper? Will the PDMS/Ecoflex liquid diffuse into the interface between acetate sheet and cellulose paper?

The surface roughness of acetate sheet was not provided by the sheet manufacturer. However, it is safe to assume that they have a mirror-like surface finish since these sheets are manufactured by thermal extrusion using roll to roll manufacturing. Additionally, we bonded the droplet generator chip with a blank PDMS slab to enclose the open channels and any surface roughness from acetate sheet would have caused leakage or lack of bonding.

While we did not quantify the adhesion forces between wet paper mold and acetate film, adhesion was always strong enough to withhold the peeling-off operation. Throughout our prototyping experiments, we did not observe diffusion of PDMS/ecoflex in between the wet paper and acetate sheet. However, the wetting properties of the glue can be an important factor which may cause such malfunction. We have not explored this experimentally mainly because finding different adhesive backed paper off-the-shelf is challenging.

We introduced the above points in the discussion, so the readers and potential end-users are aware of possible failure modes.

  1. How the surface roughness of molded microfluidic channel influence the liquid inside?

Surface roughness is a key parameter that will affect device performance in certain applications. An example of such application is monolayer of endothelial cells inside a microfluidic channel, which is often used in organ-on-chip assays. If the surface is rough, it directly affects adhesion and distribution of cells on the surface. Instead of a monolayer and uniform coverage, cellular aggregates are formed. Our follow up publication will focus on this aspect more and how we utilize the proposed micromodling method to form thin blood capillaries.

We added this to the results and discussion section.

  1. Indicate the unit of Ra and Rq in Figure 5g and h.

Both Ra and Rq were measured in microns. We added axis units in the corresponding plots.

  1. Figure 1 in page 8 should be Figure 7 and Figure 7 in page 11 should be Figure 8.

Thanks. We reconfirmed figure references in the revision.